# Malignant Bowel Occlusion: An Update on Current Available Treatments

**DOI:** 10.3390/cancers17091522

**Published:** 2025-04-30

**Authors:** Benedetto Neri, Nicolò Citterio, Sara Concetta Schiavone, Dario Biasutto, Roberta Rea, Margareth Martino, Francesco Maria Di Matteo

**Affiliations:** 1Therapeutic GI Endoscopy Unit, Fondazione Policlinico Universitario Campus Bio-Medico, 00128 Rome, Italy; benedettoneri@gmail.com (B.N.); citterion@gmail.com (N.C.); d.biasutto@policlinicocampus.it (D.B.); r.rea@policlinicocampus.it (R.R.); m.martino@policlinicocampus.it (M.M.); 2Gastroenterology Unit, Department of Systems Medicine, University ‘Tor Vergata’ of Rome, 00133 Rome, Italy; saraschiavone27@gmail.com

**Keywords:** malignant bowel occlusion, malignant colonic occlusion, malignant small bowel occlusion, MBO, MCO, MSBO, EUS-guided anastomosis, EUS-EC

## Abstract

Malignant bowel obstruction (MBO) occurs in patients with advanced malignancy, leading to chemotherapy interruption and invasive interventions. Both surgical and non-surgical treatments are available, including medical, endoscopic, and percutaneous approaches. Surgery is the treatment of choice for MBO; however, half of the patients are poor surgical candidates. Therefore, recent multidisciplinary recommendations have suggested considering less invasive interventions. Medical therapy and percutaneous approaches are effective in symptom relief but do not allow the resumption of an oral diet and thus, of oncological treatment. The role of endoscopic treatment of MBO is growing, as supported by the increase in the indications for colonic intraluminal stenting. The development of endoscopic ultrasound-guided anastomotic techniques may lead to the widening of the indications to endoscopic treatment of MBO, as they also allow an effective treatment for small bowel obstructions. The introduction of new interventional endoscopic techniques will increase the availability of minimally invasive solutions for MBO.

## 1. Introduction

Malignant bowel obstruction (MBO) is a critical complication in oncological patients, associated with high morbidity and mortality. The need to interrupt chemotherapy treatment in advanced-stage tumors in cases of MBO occurrence is one of the main implications of this condition [1]. In the International Conference on MBO and Clinical Protocol Committee, MBO is defined as (a) clinical evidence of intestinal obstruction (anamnesis, physical examination, imaging); (b) intestinal obstruction distal to the Treitz ligament; and (c) diagnosis of incurable intra-abdominal malignancy or a non-intra-abdominal primary cancer with clear abdominal carcinomatosis [1].

In retrospective studies, 10–28% of patients with gastrointestinal malignancy and approximately 50% of patients affected by ovarian cancer were shown to develop MBO [2,3]. Considering the epidemiology of gastrointestinal (GI) and gynecological malignancies, and the increasing availability of oncological therapies, the treatment of MBO is an issue that can possibly involve a relevant proportion of oncological patients.

Current treatments available for MBO include surgical and non-surgical interventions, consisting of medical, endoscopic, and interventional radiology approaches [4]. Surgery has long been considered the main treatment of MBO. However, it is not feasible in almost 50% of patients, mostly due to poor performance status [3]. Recently, MASCC multidisciplinary recommendations have suggested considering less invasive interventions instead of palliative surgery, which is related to a high risk of complications, for the management of MBO in patients with advanced malignancy [5].

In the context of biliopancreatic malignancy, the endoscopic approach is considered as the gold standard for treating malignant biliary tract obstruction (MBTO), having largely replaced surgery. Endoscopic retrograde cholangiopancreatography (ERCP) is still the pivotal biliary drainage procedure [6], even if endoscopic ultrasound-guided biliary drainage (EUS-BD) has proven its comparable effectiveness and safety [7,8]. These recent evidences have proved that this technique can replace ERCP, particularly in special situations such as altered anatomy and duodenal obstruction.

As for MBTO, endoscopic management is also becoming increasingly important for MBO. Indeed, the last update of the European guidelines on the use of colonic stents for malignant colonic obstruction (MCO) has expanded the indication for this technique. Moreover, recent retrospective studies have assessed the effectiveness of endoscopic ultrasound-guided entero-colostomy (EUS-EC) with a lumen apposing metal stent (LAMS) in patients with MBO who are not eligible for surgery, with encouraging results [9,10,11].

The present review aims to summarize the current evidence on the management of patients with MBO, with particular focus on the most recent endoscopic treatments available.

## 2. Methods

The PubMed and Scopus databases were consulted using the following search terms: ‘malignant bowel obstruction,’ ‘MBO’, ‘malignant small bowel obstruction’, ‘MSBO’, ‘malignant colonic obstruction’, ‘MCO’, individually or in combination with ‘surgery’, ‘endoscopy’, ‘interventional radiology’, ‘treatment’, ‘management’, ‘study’. The search included all compatible full-text papers published in English. No publication date restrictions were selected. Aiming to provide a comprehensive review on the management of MBO, the efficacy and safety of currently available alternative treatments have been categorized into medical, surgery, endoscopy, and interventional radiology.

## 3. Malignant Bowel Obstruction

MBO is predominantly related to advanced ovarian and colorectal cancers; nonetheless, it may also occur in other abdominal and non-abdominal malignancies [2,5]. Peritoneal carcinomatosis is the main cause of intestinal involvement in advanced metastatic cancer. Usually, it occurs as diffuse metastases, even though in 10% of cases an isolated gastrointestinal metastasis is observed [12]. Breast cancer and melanoma are the most common non-gastrointestinal malignancies causing MBO, which usually occurs in the latest stages of the disease [13]. A recent review by Pujara et al. [14] reported that, in a cohort of 334 patients with intestinal obstruction, the main causes were advanced malignancy (68%), adherence (20%), and unclear etiology (12%).

The pathogenesis of MBO includes mechanisms that can be distinguished into mechanical and functional [2,5,15].

Mechanical obstruction can be classified in the following circumstances:−Extrinsic luminal occlusion, due to growth or recurrence of the primary tumor, mesenteric and omental masses, intra-abdominal adhesions, or post-irradiation fibrosis that may determine bowel compression;−Intra-luminal occlusion, due to tumor growth from intestinal lumen;−Intra-mural bowel occlusion, determined by intestinal linitis plastica, a tumor growing inside the intestinal wall determining poor motility.

Functional mechanisms of MBO include the following:
−Motility disorders determined by tumor infiltration of the mesentery or intestinal wall and nerves or neoplastic involvement of the coeliac plexus;−Motility disorders such as paraneoplastic neuropathy (mainly in patients with lung malignancy), chronic intestinal pseudo-obstruction, or paraneoplastic pseudo-obstruction.

Furthermore, non-malignant factors may contribute to intestinal obstruction, such as constipation/fecal impaction, pharmacological (i.e., opioids, intra-peritoneal chemotherapy), fibrosis, and adhesions due to surgery and/or radiotherapy [5].

MBO includes MCO and MSBO. Bowel obstruction may be partial or complete and can present with single or multiple localizations [16]. MBO affects the small bowel only in more than half of cases (61%) and the large bowel in isolation in 1/3 of patients [16]. Both the small bowel and the colon are affected by MBO in 20% of patients [16].

MBO leads to a reduction or absence of peristalsis hesitating in bowel distension [15,17]. The accumulation of content in the bowel lumen triggers an inflammatory response characterized by intestinal edema and hyperemia, driven by massive release of inflammatory mediators (i.e., prostaglandins, vasoactive intestinal polypeptide and nociceptive mediators). Bacterial overgrowth and translocation are deeply involved in symptoms occurrence [5].

Clinically, MBO is characterized by a gradual worsening of symptoms, such as cramping, abdominal pain, and distension related to nausea, fecaloid vomiting and retention of gasses and stools [5]. MBO onset can also be insidious, requiring a time span of several weeks to develop from mild to severe.

Furthermore, obstruction symptoms can be intermittent, with spontaneous resolution occurring in up to 36% of patients with inoperable MBO. However, the rate of obstruction recurrence exceeds 60% of cases [18].

When MBO is suspected, plain abdomen RX is usually the first diagnostic procedure performed, followed by an abdominal contrast-enhanced CT scan which allows the localization of the site of the occlusion [1,12,19]. Even though MRI is feasible and characterized by high sensitivity, specificity, and accuracy (95%, 100%, and 96%, respectively), it is rarely used for diagnosis since is it rarely adopted in an urgency setting [20].

Once the diagnosis of MBO has been made, patients must be stabilized as soon as possible and started on palliative treatment, which must be determined based on the patient’s characteristics. Indeed, treatment goals should be decided among patients, caregivers, oncologists, surgeons, and endoscopists.

Initial basic medical management involves nasogastric tube placement to decompress the GI tract and to prevent possible aspiration pneumonia coupled with the infusion of IV fluids and electrolytes to allow patient hydration and treat possible plasmatic electrolytes alterations.

The therapeutic aim in the management of MBO is to alleviate occlusive symptoms and, when possible, to resolve the obstruction. This is mandatory in order to allow the patient to return to cancer treatment. The risks associated with possible available procedures, the prognosis, the overall performance status, and the main comorbidities have all to be considered to choose the optimal strategy [5].

The management of patients with MBO may involve conservative treatments, such as drug therapies, and interventional procedures [4,21].

## 4. Medical Treatment

Medical therapy aims to alleviate symptoms such as abdominal pain, nausea, and vomiting. Drugs used in this context include analgesics (e.g., opioids), anti-emetics (e.g., 5-HT3 inhibitors), pro-kinetics (e.g., dopamine antagonists), anticholinergics (e.g., hyoscine butylbromide), and anti-secretory agents (e.g., somatostatin analogs). Somatostatin analogs (i.e., octreotide and lanreotide) are among the most effective therapies for the treatment of nausea and vomiting, showing the highest level of evidence in the literature [5]. These drugs can be administered subcutaneously or as an intravenous continuous infusion. For abdominal pain, corticosteroids and anticholingerics can be used [5]. However, in approximately 80% of cases, this conservative management strategy alone fails [22].

Therefore, medical treatment is usually only a part of the therapeutic strategy when managing patients with MBO.

Interventional treatments can be divided into surgical and non-surgical procedures, which comprise percutaneous techniques and endoscopic treatments.

## 5. Surgery

MBO is the most common indication for palliative surgical consultation [23]. Palliative surgery can be successful in the case of an MBO with a single obstruction site, but is less effective in presence of intra-abdominal carcinomatosis [4,15]. Patients with MBO secondary to abdominal metastases may have more severe symptoms that are not responsive to medical therapy [14]. In this scenario, the surgical decision is very arduous. Indeed, while surgical intervention could solve the obstruction syndrome, on the other hand, patients with peritoneal metastases may have poor prognosis. This is not only related to the underlying malignancy but is also due to malnutrition and concomitant diseases, which often make these patients frail surgical candidates.

Several types of surgical interventions for MBO palliation have been reported, including ostomy (colostomy, ileostomy, or jejunostomy), intestinal resection and/or bypass (entero-enterostomy, entero-colostomy, or colo-colostomy), and lysis of malignant adhesions [24,25,26,27,28,29,30,31,32,33,34,35,36,37,38].

The current evidence is characterized by significant heterogeneity as far as the type of procedures performed, the study populations, and the outcomes analyzed are concerned [2]. Therefore, it is difficult to obtain clear indications regarding which surgical procedure is to be preferred in the different possible scenarios. Surgical interventions in a single-site obstruction may involve open surgery with bypass or ostomy [21].

Most frequently, MSBO is secondary to a condition of diffuse carcinomatosis, for which a more conservative approach is preferable to surgery whenever the strangulation of intestinal loops is not present [17]. When needed, possible surgical interventions include small bowel resection with anastomosis, bypass, and terminal ostomy [39]. Small bowel ostomy is usually preferred in case of an obstruction site distal enough to allow gastrointestinal autonomy, thus lowering the risk of short bowel syndrome, or in case of high risk of anastomotic complications [23]. Differently, small bowel resection is suggested in presence of a single carcinomatosis nodule of primitive small bowel malignancy, which, however, are relatively infrequent conditions. MSBO with multiple obstruction sites represents a real challenge for surgeons, due to patients’ frailty, the frequently concomitant state of cachexia and sarcopenia, and the advanced stage of the malignancy. For these reasons, intestinal bypass without resection is the preferred treatments in patients with disseminated malignancy, as suggested by a UK prospective study, reporting data that up to 31.9% of patients with MSBO due to carcinomatosis underwent an intestinal bypass [40].

In the case of MCO, conservative management is not preferred, taking into account the high risk of perforation and death. In emergency scenarios, such as acute malignant large bowel obstruction, surgery is mandatory, particularly in patients with right-sided colon cancer. The most common surgical procedure is ostomy, including cecostomy or colostomy, which are to be preferred over primary resection and anastomosis or bypass [25,39]. In a multicenter prospective study including 205 patients, the most common intervention in case of right-sided colon cancer was resection with ileostomy, which was performed in up to 74.4% of cases [40]. However, in presence of diffuse carcinomatosis, bypass without resection was the most frequent procedure (31.9%). In patients with single primary tumor obstruction, resection of the tumor should be considered the intervention of choice, with regard to higher chances of oncological radicality and longer survival, wherever possible [41].

In contrast, in acute left-sided MCO, emergency surgery is not suggested as the treatment of choice. Indeed, the mortality and morbidity rates for emergency surgery in such condition range between 15 and 34% and 32–64%, respectively, being significantly higher than those of elective surgery [42]. Furthermore, in a recent meta-analysis, emergency surgery with primary anastomosis in patients with left-sided MCO was associated with poorer outcomes in terms of a higher risk of permanent ostomy, intra-hospital morbidity, and mortality [43]. Therefore, when possible, strategies allowing the bridging the patient to elective surgery, including transanal drainage or colonic stent placement, should be preferred.

Overall, a systematic review including 17 studies [25] reported that surgery allowed resolution of obstructive symptoms in between 32% and 100% of patients. Furthermore, 45–75% of patients were able to resume oral feeding after surgery and 34–87% could be discharged at home. However, high rates of post-operative mortality (32%) and complications (44%) were also reported [25]. Entero-cutaneous fistula, wound infection and dehiscence, early recurrence of the obstruction, high-output ostomy, myocardial infarction, cardiovascular failure, deep vein thrombosis, pulmonary embolism, pulmonary infections, anastomotic leak, and infections were among the reported complications. However, only 2–15% of patients underwent additional surgery to address complications. Overall, the authors reported that the re-obstruction rate was up to 47% and only 32–71% of patients remained asymptomatic and were able to tolerate oral feeding 60 days after surgery.

In another systematic review, Cousins et al. [2] reported post-operative re-obstruction rates ranging between 10% and 63% and post-surgical morbidity and mortality rates of 87% and 32%, respectively. Ripamonti et al. [15] also reported a high 30-day mortality rate (ranging from 21% to 40%) and a high complications rate (20–40%).

In the light shed by these data, there are no clear indications on how to identify patients who have the highest odds of successful surgical palliation of MBO. The presence of palpable intra-abdominal mass, ascites, intra-abdominal carcinomatosis (vs. isolated local recurrence), several obstruction sites, advanced malignancy, and poor overall clinical status have all been suggested as negative prognostic factors for palliative surgery success [1].

A recent multicenter retrospective study including 70 patients affected by MBO [3] reported a mortality rate at 30-day of approximately 20%. Furthermore, the authors reported that older age and macroscopically visible metastatic foci were associated with a higher 30-day mortality at multivariable analysis, with OR of 1.21 [1.07–1.37] and 49.61 [3.25–758.06], respectively.

Given the high post-operative mortality rate and the frailty of MBO patients, who often have a poor prognosis, the MASCC MBO study group recommends less invasive surgical interventions and suggests that palliative surgical interventions may be considered only in a highly select population [5].

## 6. Percutaneous Techniques

Less than half (36–49%) of MBOs resolve with medical management alone. However, recurrence rates are as high as 72% [18,44]. As mentioned above, surgical management is feasible only in approximately 50% of patients [2,3], who can also benefit from less invasive interventions, such as nasogastric tube placement. A nasogastric tube is only recommended for a very short time period, because of discomfort associated with the presence of the device and related complications, including nasal necrosis, esophageal and gastric erosions, and aspiration [45].

For longer periods, patients with MBO may benefit from percutaneous procedures, which can be both radiological and endoscopic. These include “venting” percutaneous transabdominal gastrostomy (PTAG), percutaneous transesophageal gastric (PTEG) catheter placement, percutaneous jejunostomy, and percutaneous cecostomy/colostomy (PC). Consensus guidelines and MASCC recommendations suggest the use of percutaneous techniques in patients who do not respond to medical antiemetic therapy, are unsuitable for surgery, or with very poor prognosis (i.e., life expectancy of less than 30 days) [5,15].

PTAG consists of the placement of a decompression catheter, either radiologically or endoscopically (i.e., percutaneous endoscopic gastrostomy, PEG). In a recent systematic review of 25 studies by Thampy et al. [46], comprising 1194 patients, the authors demonstrated that insertion of a venting gastrostomy has high technical and clinical success rates (91% and 92%, respectively), with an overall major and minor complication rates of 1.9% and 19.8%, respectively. Not all of the studies considered assessed the ability to re-feed orally; however, in those reported, oral feeding was feasible in 84% (422/504) of cases. The reported median patients’ survival ranged from 17 to 74 days, while the mean survival ranged from 35 to 147 days. Contraindications to PTAG catheters are mainly related to anatomical conditions that may limit transabdominal gastric access. These include concomitant ascites, altered anatomy due to previous surgery, interpositioning of small bowel or colonic loops, intra-abdominal carcinomatosis, and gastric cancer (primary cancer or metastases) affecting the anterior gastric wall [12,29,47,48]. In patients with ascites, altered anatomy, or other conditions that can make PTAG placement difficult, percutaneous transesophageal gastric catheter insertion can be considered [5,49,50,51,52,53,54,55,56]. PTEG placement requires an inflated esophageal dilatation balloon, inserted via a snare under ultrasound guidance. Through a puncture, a guidewire is passed within the balloon and then both are advanced into the stomach. Eventually, by adopting the “Seldinger technique”, the catheter is placed in the gastric fundus or body [57]. In a retrospective study conducted on 385 patients by Litwin et al. [48], procedural technical success rates were 95.5% and 97.6% for PTAG and PTEG, respectively. Furthermore, the authors observed similar procedural complication rates for both procedures: 13.9% for PTAG and 22.5% for PTEG [48].

Percutaneous jejunostomy is an alternative technique to PEG and transgastric jejunostomy [58], particularly when the stomach has been removed or is inaccessible [59,60]. To date, percutaneous jejunostomy is mostly carried out endoscopically, but radiological techniques have also been reported [59,61].

Percutaneous cecostomy or colostomy is a viable alternative to open or laparoscopic surgical cecostomy to treat MCO [62,63]. In a retrospective study [64], image-guided percutaneous colostomy was reported as a safe and effective technique for management of large bowel malignant obstruction, with a pain relief rate of 89%. Schwingel et al. [65] recently reported the case of a 67-year-old patient with malignant obstruction of the ascending colon, in which decompression was successfully achieved via percutaneous endoscopic colostomy by using a lumen-apposing metal stent (LAMS), with full bowel decompression and resumption of oral diet after only 2 days.

## 7. Endoscopic Treatments

### Colonic Stenting

Current European guidelines recommend colonic stenting using self-expandable metal stents (SEMS) in patients with clinically relevant bowel obstruction and in those with radiological suspicion of MCO without signs of perforation [66]. Endoscopic decompression historically has the aim of converting emergency surgery into elective, one-stage surgery, thus reducing the adverse events risk and the need for temporary stoma [67]. Currently, indications to SEMS placement are not limited to “bridge to surgery” anymore. Indeed, SEMS placement is also suggested in other situations such as inoperable malignancy, as symptom palliation to improve patient’s quality of life [68], and in extra-colic neoplasm determining MBO [69]. Colonic stenting can be achieved either by using through the scope (TTS) or the over-the-wire (OTW) techniques and requires the use of both endoscopy and fluoroscopy [66]. The use of uncovered SEMSs, according to the European guidelines, is recommended in both in curative and palliative settings [66]. Their use is indeed associated with lower complication rates when compared to covered SEMSs (risk ratio [RR] 0.57). These include less tumor overgrowth (RR 0.29), less SEMS migration (RR 0.29), and longer patency (mean duration 18 months), although the risk of tumor ingrowth is higher (RR 4.53) [70].

## 8. Colonic Stenting in Resectable Patients: A Bridge to Elective Surgery

Multiple professional societies recommend SEMS as a bridge to surgery in patients with potentially curable left-sided obstructing colon cancer. SEMS placement (Figure 1) allows the performance of elective surgery, preoperative bowel cleansing, and full preoperative staging, thus optimizing the odds of performing a one-stage curative surgery rather than an upfront emergency one [67,71].

A recent meta-analysis of randomized controlled trials (RCTs) [72] compared SEMS implantation versus emergency resection for the management of malignant left-sided colorectal obstruction. The paper reported a lower overall incidence of complications (RR = 0.787, *p* = 0.004), incision infection rate (RR = 0.472, *p =* 0.003), permanent stoma rate (RR = 0.499, *p =* 0.0001), overall stoma rate (RR = 0.520, *p* = 0.0001), and higher primary anastomosis rate (RR = 0.472, 95% CI: 0.286–0.7 77, *p =* 0.003) in the SEMS group when compared to patients undergoing emergency resection [72].

McHugh et al. [73], in a network meta-analysis, also reported that colonic stenting was associated with a reduction in the permanent stoma rate when compared to emergency resection (RR 0.57; 95% CI 0.33–0.79). Colonic stenting also frequently allowed more minimally invasive surgery (RR 4.10; 95% CI 1.45–13.13) and was associated with lower morbidity (RR 0.58; 95% CI 0.35–0.86). Moreover, a pairwise analysis of primary anastomosis rates showed that this was increased by SEMS adoption (RR 1.40; 95% CI 1.31–1.49) rather than emergency resection. Overall, SEMS implantation is a safe strategy for the treatment of left-sided MCO, with long-term advantages over emergency resection, including a lower permanent stoma risk and a significantly higher proportion of primary anastomoses.

Although the evidence regarding the utility of SEMS in different settings for left-sided MCO is growing, their adoption for the treatment of right-sided or proximal (i.e., proximal to the splenic flexure) neoplastic obstructions still raises concerns. This is mainly due to the technical challenges of SEMS insertion in those areas. Meta-analysis data on SEMS as a bridge to surgery in right-sided MCO reported a high stent success rate (OR 0.92, 95% CI, 0.87 to 0.95) and a low perforation rate (OR 0.03, 95% CI 0.01–0.06) [74], coupled with reductions in post-operative complications (OR = 0.78; 95% CI 0.66–0.92) and mortality (OR = 0.51; 95% CI 0.28–0.92) when compared to emergency resection [75]. However, no randomized controlled trials are available, suggesting the need for further research in this field.

## 9. Palliative Colonic Stenting in Non-Resectable Patients

For patients with MCO that are not eligible for surgical resection, SEMS placement is a useful alternative that allows both comfort, by avoiding a stoma, and, potentially, palliative therapies. SEMS placement and palliative surgery for MCO were compared in four systematic reviews and/or meta-analyses (including randomized and non-randomized studies) that reported a technical success rate of stent placement ranging from 88% to 100%, with a shorter hospitalization duration with respect to surgery [76,77,78,79]. Overall, morbidity was comparable between the two groups [77,78]. However, two meta-analyses reported a higher frequency of early complications in the surgery group, while late complications were more common in the SEMS group [78,79]. The most common stent-related complication was perforation [76,77,78,79].

SEMS placement is mainly used also in the palliative setting for left-sided MCO. However, it could be considered also for the treatment of proximal colon malignant obstructions, even though this is known to be difficult. Therefore, when available, this procedure should be performed by an operator experienced in interventional endoscopy [66]. A recent multicenter study [80] compared the effectiveness of SEMS placement in left vs. right-sided MCO. The reported clinical success rates were 97.1% in patients with right-sided malignancy and 88.2% in those with left-sided MCO. Complications occurred in 10.1% of patients with right-sided MCO and in 19.9% of those with left-sided MCO, and included stent migration, tumor ingrowth, overgrowth, perforation, bacteremia/fever, and bleeding. Overall, SEMS seems a viable option also in patients with right-sided MCO, although more data are needed [80].

Regarding MCO caused by extracolonic malignancy, that is due to extrinsic compression, mesenteric infiltration, and dysmotility rather than space-occupying growth caused by colorectal cancers [71], the role of colonic stents is controversial and it has been studied mainly retrospectively. Two studies, involving 44 and 60 patients with extracolonic malignancy, reported no differences in technical, clinical, and AEs rates in patients undergoing palliative SEMS placement with extracolonic or primary colorectal malignancy [81,82]. A recent meta-analysis, including eight non-randomized studies, assessed the safety and efficacy of SEMS placement when treating extracolonic malignancy when compared to intracolonic malignancy [83]. In this paper, a higher risk of technical failure (RR 2.92; 1.13–7.54) and of clinical failure (RR 2.88; 1.58–2.52) for extracolonic malignancy, with a higher risk of perforation (RR 3.22; 1.44–7.19; *p* = 0.004), were reported. When compared to emergency decompressive surgery, SEMS placement for extracolonic malignancy has shown fewer complications, but lower clinical efficacy (technical success, 73.9% vs. 94.2%, *p* = 0.001; clinical success, 54.1% vs. 75.4%, *p* = 0.005) [84]. Colonic stents are a valid alternative for the treatment of patients with extracolonic malignancy who are poor surgical candidates, even though this procedure is more technically difficult, the clinical success rates are slightly lower, and the complication rates are not negligible.

## 10. Endoscopic Ultrasound-Guided Entero-Colostomy with LAMS

In the last few years, a novel application of EUS-guided LAMS placement to create an entero-colonic bypass in patients with MBO has been introduced, showing encouraging results.

The first case reports are from 2019. Mir et al. [85] reported the case of a 72-year-old patient with advanced metastatic pancreatic adenocarcinoma, who presented with recurrent MBO and was deemed unfit for surgery, who underwent EUS-guided 15 mm × 10 mm LAMS (Axios; Boston Scientific, Marlborough, MA, USA) placement from the sigmoid colon to create an entero-colostomy (i.e., LAMS positioning between sigmoid colon and a dilated bowel loop), bypassing and resolving the distal small bowel obstruction, in absence of immediate postoperative adverse events. Sooklal et al. [86] reported the case of a 43-year-old patient with an extensive surgical history, affected by metastatic undifferentiated signet-ring cells colonic adenocarcinoma with peritoneal carcinomatosis, presenting with recurrent small bowel obstruction. Due to the high risks and the technical limitations related to both surgical and interventional radiology treatment, the patient’s management was endoscopic. From the second duodenal portion, under endosonographic guidance, a 15 mm × 10 mm LAMS (Axios; Boston Scientific, Marlborough, MA, USA) was deployed between the duodenum itself and an adjacent segment of colon, creating an enterocolostomy, followed by clinical resolution of the obstruction.

Most recently, three retrospective studies on this topic were published. Jonica et al. [9] enrolled 10 patients with MSBO at 3 tertiary endoscopy U.S. centers, considered at high risk for surgery, who underwent EUS-guided LAMS placement to create an enterocolostomy for palliation of acute obstruction. EUS-guided entero-colostomy (EUS-EC) was performed transanally in all patients by experienced endosonographers. After EUS-guided identification of a dilated small bowel loop proximal to the stenosis, a cautery-enhanced LAMS (15 × 10 mm or 20 × 10 mm AXIOS Stent and Electrocautery Enhanced Delivery System; Boston Scientific, Marlborough, MA, USA) was deployed under endosonographic and fluoroscopic visualization, creating an anastomosis between the colon and small bowel. The reported technical success (defined as the correct deployment of the LAMS between the colon and the small bowel) was 80%, while clinical success (defined as the resolution of the obstruction for at least 14 days after the procedure) was 70%, with only one major AE (10%), which was aspiration.

Mitsuhashi et al. [10] included 26 patients from 2 tertiary referral medical centers who underwent the EUS-EC for the management of MSBO. Before the EUS-EC procedure, all patients failed other methods of bowel decompression. LAMS were either 15 × 10 mm or 20 × 10 mm AXIOS stent loaded on an electrocautery-enhanced delivery system (AXIOS; Boston Scientific, Marlborough, MA, USA). The used approaches were both entero-colonic (i.e., echoendoscope advanced as deeply as possible into the small bowel, with subsequent LAMS deployment, typically performed freehand) and colo-enteric (i.e., echoendoscope advanced through rectum with subsequent cautery-enhanced LAMS deployment, under EUS and fluoroscopic guidance).

The technical success (defined as the successful deployment of the LAMS between the colon and a small bowel proximal obstruction site) rate and clinical success (defined as the resolution of obstructive symptoms) rates were, respectively, 100% and 92.3%. AE occurred in four (15.4%) patients and included bleeding, diarrhea, and post-procedure sepsis.

Neri et al. [11] enrolled 12 consecutive patients at 4 different tertiary referral centers across Europe, who were not candidates of surgery and/or colonic SEMS placement, treated by EUS-EC with LAMS for MBO. Among the patients enrolled, MBO included both MCO and MSBO due to primary tumors or metastases. EUS-EC was performed using a transanal approach in all patients. A cautery-enhanced LAMS (15 × 10 mm, 20 × 10 mm, 16 × 20 mm; Hot AXIOS Stent and Electrocautery-Enhanced Delivery System; Boston Scientific, Marlborough, MA, USA; Hot SPAXUS Stent; Taewoong Medical Co., Gimpo, Republic of Korea) was directly deployed under EUS and fluoroscopic visualization to create an anastomosis between the colon and a dilated small bowel or colonic segment proximal to the stenosis (ileo-colostomy and colo-colostomy, respectively) (Figure 2). In this study, a slightly different technique has been adopted, since all LAMS have been released with a freehand technique. The reported technical success (i.e., ability to deploy the LAMS between the colon and the colonic or small-bowel target) and clinical success (i.e., resolution of MIO for at least 2 weeks after the procedure) rates were 100% and 83.3%, respectively. No LAMS misdeployments or other intraprocedural AEs were reported. Three patients (25%) developed severe post-procedural complications, all septic, which led to the death of two. Differently from the studies of Jonica and Mitsuhashi, in this study, the procedure and patients’ outcomes have been compared between ileo-colonic and colo-colonic anastomoses and appear statistically comparable. Though limited by the small study population, the study suggests that, when treating MBO, it is not relevant which target proximal to the obstruction is chosen, as long as it is the easiest and the safest.

Overall, EUS-EC seems to be associated with high technical and clinical success rates, with reasonably low AEs, when performed by skilled endoscopists. This procedure is therefore an alternative opportunity for the minimally invasive treatment of patients with MSBO that cannot be managed with surgery or interventional radiologic or other endoscopic approaches, due to patient-specific characteristics or anatomic restraints.

## 11. Expert Opinion

The management of MBO is a tough challenge for all the clinicians involved. This is due to several reasons, including the emergency setting, the different possible characteristics of both the occlusion and the patients, the frailty of patients and, last but not least, the ethical aspects. Therefore, teamwork and collaboration among the different specialists involved is crucial to aim for the best possible strategy. A proposed algorithm summarizing the available interventional treatments of MBO is reported in Figure 3.

The availability of a growing number of interventional procedures that allow both symptom resolution and the ability to resume oral nutrition has limited the role of medical therapy to the stabilize the patients’ conditions before other treatments or to treat end-stage patients. Nonetheless, an optimal medical therapy is crucial, as it can maximize the odds of subsequent more effective palliative treatments.

When considering interventional procedures, a difficult balance should be reached between short- and medium-term efficacy and safety. The latter aspect is of particular concern, taking into account that most patients have to be treated in poor conditions due to the advanced malignancy by which are affected. Therefore, in recent years, a shift has been observed from more aggressive surgical strategies involving resection of the stenotic intestinal tracts towards interventions aimed at allowing the restoration of the bowel continuity.

With this in mind, the role of percutaneous techniques, both endoscopic and radiologic, is confined to patients with non-mechanical obstruction, characterized by intermittent symptoms and spontaneous resolution, and with very short-term prognosis. These techniques, indeed, allow symptom relief by reducing the intraluminal pressure but do not guarantee the restoration of the continuity of the GI tract. Therefore, the gain in terms of oral diet resumption may be limited. For this reason, these procedures are usually reserved for patients with MSBO in very advanced stages, as the only advantage provided is the resolution of main obstructive symptoms, such as pain and vomit. Indeed, when dealing with MCO, multiple therapeutic options are available and percutaneous techniques (i.e., cecostomy) are rarely adopted.

Surgery is still considered the standard of care when managing MBO. Therefore, when facing such situations, a surgical consultation is mandatory in order to arrange the best therapeutic strategy on a patient-to-patient basis and according to local availability. Overall, the safest and simplest surgical intervention should be preferred. Indeed, surgery is burdened by a very high post-operative mortality rate (up to 20%) [3]. Therefore, more complicated operations involving intestinal resection should be reserved for a highly select number of patients [5], such as those with a single malignant stenosis and candidates for further oncological treatments. Otherwise, ostomies and bypasses may be preferred in case they do not lead to short bowel syndrome. Another important shift in the surgical management of MBO and, particularly, of MCO, is that efforts are suggested to avoid emergency surgery. Given this, the wider use of colonic SEMS or decompression tubes is suggested that, by lowering the risks of perforation due to the obstruction, allow to perform elective interventions, characterized by higher efficacy and lower morbidity and mortality.

Endoscopic management of MBO in recent years is assuming a more relevant role, as supported by the widened indications to colonic intraluminal stenting in the latest update of the European guidelines [66]. However, until recently, the endoscopic approach could be considered as a useful tool only for patients with MCO. In this population, the use of SEMS is currently suggested not only as a bridge to surgical treatments but also as a definitive therapy in a palliative setting, particularly for left-sided MCO. Indeed, when compared to surgery, SEMS placement has a significant advantage in terms of safety but has lower long-term efficacy rates. However, it has to be considered that, when managing advanced oncological patients, the long term is not always very long and that even for situations in which SEMS maybe less effective, such as with extracolonic malignancies, clinical success is achieved in more than half of the patients [84]. Concern remains regarding right-sided MCO, in which SEMS placement is more demanding and considerable experience is required in order to avoid complications. Even in referral centers, the difficulty of SEMS placement in the right colon may be due to sigmoid and descending colon angulations, carcinomatosis, and ascites that can prevent the success of the procedure. Moreover, in this situation, negative oncological implications have been suggested [66]. The availability of both TTS and OTW techniques for SEMS placement and of different endoscopic guidewires with variable degrees of stiffness may allow the difficulties related to this procedure to be partially overcome.

These considerations led very recently to the proposal of new interventional endoscopic techniques that allow the formation of anastomoses, bypassing the obstruction sites, similarly to what was already available for the management of duodenal neoplastic obstructions. These techniques may also allow the treatment of MSBO, differently from endoluminal stenting which is indicated only for MCO. Taking into account the outcomes of the EUS-guided GEA with LAMS, in recent years, the safety and efficacy of EUS-EC have been tested in few studies [9,10,11]. In the three studies available from Europe and the US, the technical and clinical success rated of EUS-EC were all above 80% and 70%, respectively. The complication rates were not irrelevant, being up to 25%; however, they were at least comparable, if not lower, than that of surgery. This is of particular interest considering that in these studies only patients who were not candidates for surgery were enrolled and that these were the first experiences with this technique.

## 12. Future Directions

The improvement and enrichment of the oncologic therapeutic armamentarium may implicate the possibility of also treating very advanced tumors and a longer survival of patients with malignancy. This may lead to an increased possibility to treat MBO due to advanced malignancies and carcinomatosis, also in older and frailer patients. Is this scenario, the availability of less invasive approaches, allowing resolution on the obstruction and resumption of chemotherapies, is crucial. As suggested by the MASCC MBO study group, the safest surgical procedure should be preferred for the treatment of MBO [5]. Aiming for the lest invasiveness, endoscopic procedures may provide a valid alternative, not only to allow elective surgery but as definitive treatments, currently at least for MCO. From this perspective, the development of EUS-guided anastomotic techniques may offer the possibility not only to treat inoperable patients with MBO but also patients at high risk of surgery failure. Considering the challenges of such endoscopic procedures, the main limitations are represented by the low availability and lack of a standardized technique, which, however, are due to the novelty of this intervention.

In the future, more choices will be available for the treatment of complex conditions such as MBO, which will hopefully lead to better quality of life and better management, even of end-stage oncological patients. The rapid evolution of endoscopic techniques and their progressive diffusion will be of great help to achieve minimally invasive solutions to such critical conditions.

## Figures and Tables

**Figure 1 cancers-17-01522-f001:**
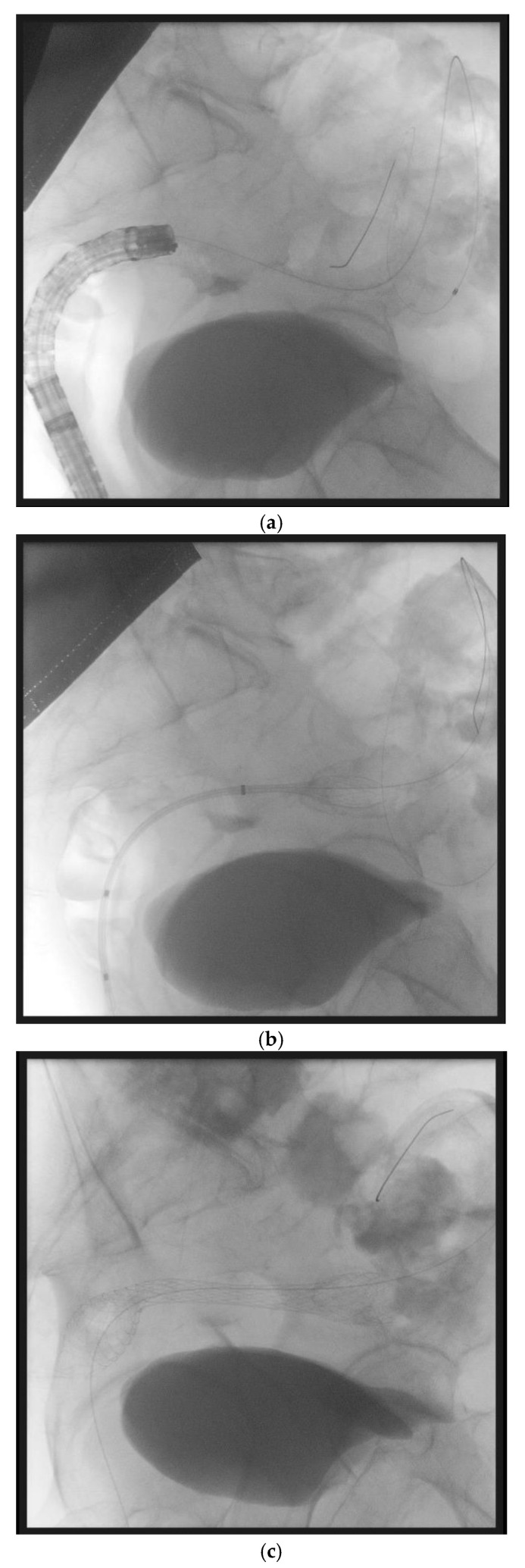
(panels **a**–**d**) Sequential steps of endoscopic placement of a self-expanding metal stent for the treatment of a left-sided malignant colonic obstruction due to metastatic colorectal cancer. In this case, an over the wire technique has been used. First, a guidewire and a Huibregtse catheter have been passed through the stenosis under endoscopic and fluoroscopic guidance (panel **a**). Then, a colonic uncovered metal stent has been placed under fluoroscopic guidance (panel **b**) and progressively released (panel **c**). Endoscopic control confirms the correct placement of the stent (panel **d**).

**Figure 2 cancers-17-01522-f002:**
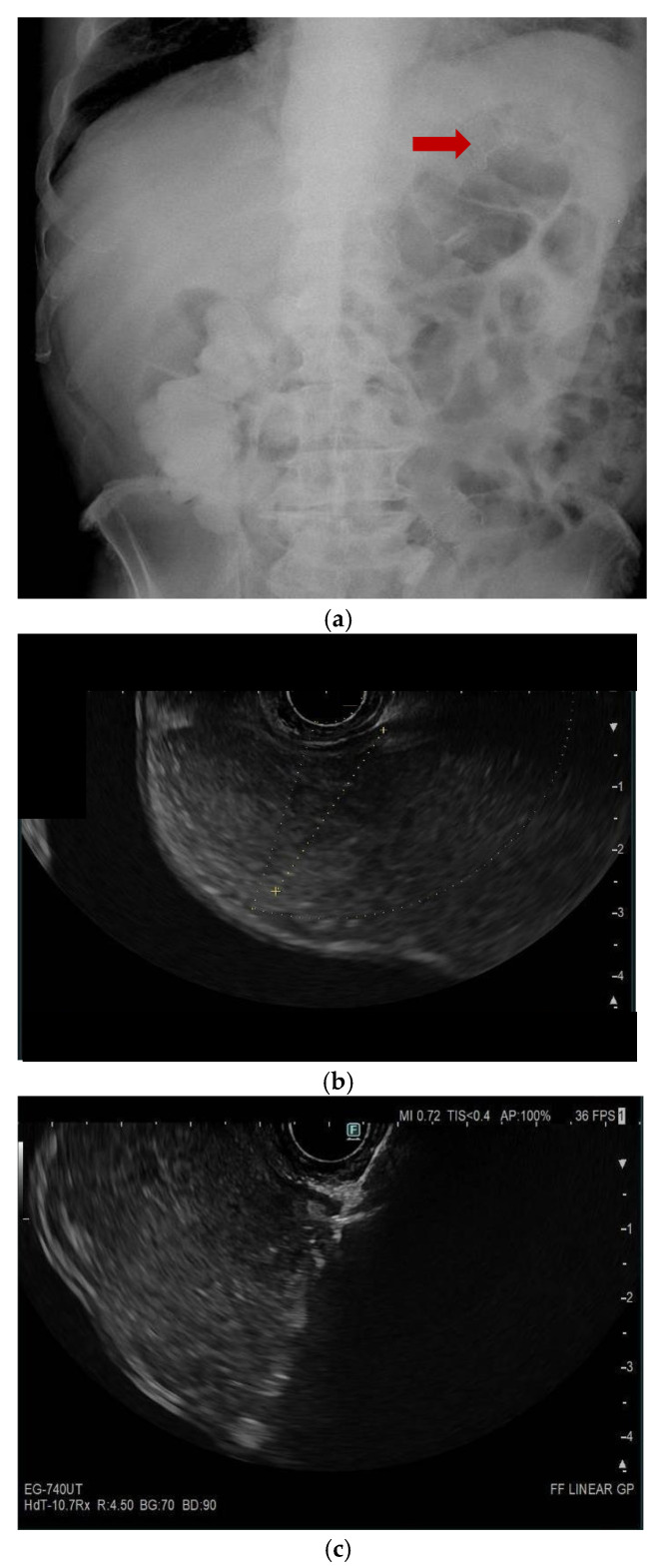
(panels **a**–**d**) Endoscopic ultrasound (EUS)-guided entero-colostomy performed in a patient with malignant colonic obstruction due to carcinosis from an advanced cholangiocarcinoma. The patients had previously undergone an EUS-guided gastroenteroanastomosis lumen-apposing metal stenting (LAMS), as shown in a plain RX study of the same day of the procedure (arrow, panel **a**). A therapeutic linear echoendoscope is advanced transanally until an adequate endosonographic window is reached (panel **b**). Then, with endosonographic and fluoroscopic guidance, the LAMS is released (panels **c**–**e**). The technical success is confirmed by passage of feces and contrast through the LAMS (panel **f**).

**Figure 3 cancers-17-01522-f003:**
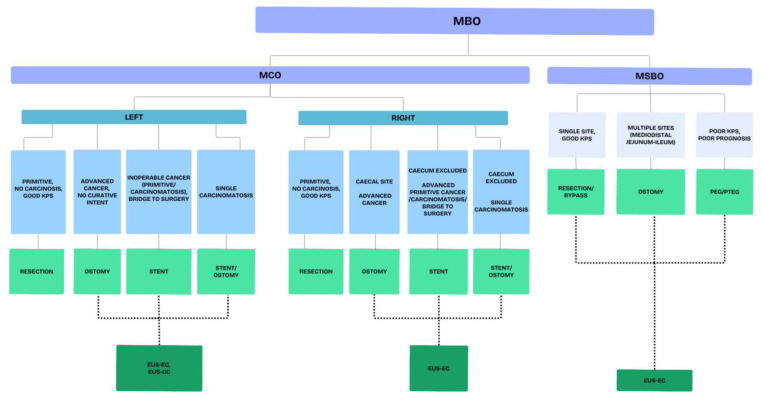
Proposed algorithm for the treatment of malignant bowel obstruction. Legends: MBO—malignant bowel obstruction; MCO—malignant colonic obstruction; MSBO—malignant small bowel obstruction; EUS-EC—endoscopic ultrasound (EUS)-guided entero-colostomy; EUS-CC—endoscopic ultrasound (EUS)-guided colo-colostomy.

## Data Availability

Data available upon reasonable request.

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
