# Peer review of "Malignant Bowel Occlusion: An Update on Current Available Treatments"

_cancers, 2025, doi:10.3390/cancers17091522_

Round 1

Reviewer 1 Report

Comments and Suggestions for Authors

The Authors present an update on current available treatments for malignant bowel obstruction. Through a thorough literature search they outline the main therapeutic interventions in this clinical setting, focussing on medical, surgery, percutaneous and endoscopic approaches. Also a Future Directions paragraph has been added which outlines possible less-invasive therapeutic scenarios through the use of EUS-guided anastomic techniques. In my opininon this manuscripts is of great clinical interest, the background is exaustive, methods are appropriate and all sections are well written, and easy to read. Finally, illustrations are of high quality and informative

-The topic is clinicaly relevant and takes into account all of the therapeutic options including medical, surgical and endoscopic. Paper is a comprehensive, critical, up-to-date review of all of the possible ways of managing this clinically challenging situation. Methodology is appropriate. Conclusions are consistent with the evidence  and the Authors address the main question they posed (i.e. thorough evaluation of therapeutical options available for malignant obstruction of the bvowel). Also,interestingly, the Authors  point out possible less invasive therapeutic scenarios in this particular clinical setting. Figures are appropriate and informative. 

I would only add a table summarizing the therapeutical options a of malignant obstruction of the bowel and their specific indications

Author Response

The Authors present an update on current available treatments for malignant bowel obstruction. Through a thorough literature search they outline the main therapeutic interventions in this clinical setting, focussing on medical, surgery, percutaneous and endoscopic approaches. Also a Future Directions paragraph has been added which outlines possible less-invasive therapeutic scenarios through the use of EUS-guided anastomic techniques. In my opininon this manuscripts is of great clinical interest, the background is exaustive, methods are appropriate and all sections are well written, and easy to read. Finally, illustrations are of high quality and informative

-The topic is clinicaly relevant and takes into account all of the therapeutic options including medical, surgical and endoscopic. Paper is a comprehensive, critical, up-to-date review of all of the possible ways of managing this clinically challenging situation. Methodology is appropriate. Conclusions are consistent with the evidence  and the Authors address the main question they posed (i.e. thorough evaluation of therapeutical options available for malignant obstruction of the bvowel). Also,interestingly, the Authors point out possible less invasive therapeutic scenarios in this particular clinical setting. Figures are appropriate and informative.

I would only add a table summarizing the therapeutical options a of malignant obstruction of the bowel and their specific indications

We wish to thank the reviewer for his comments. As requested, a figure summarizing the therapeutical interventional options for the management of malignant bowel obstruction according to specific indications has been added (Figure 3).

Reviewer 2 Report

Comments and Suggestions for Authors

It is a good review of the literature that captures all the important aspects of occlusions. Table 1 can be removed, it is not relevant.

Author Response

It is a good review of the literature that captures all the important aspects of occlusions. Table 1 can be removed, it is not relevant.

We wish to thank the reviewer for this comment. As requested, Table 1 has been removed.

Reviewer 3 Report

Comments and Suggestions for Authors

As written by the authors in lines 79-80 the aim of the submitted paper „is to give an overview of the currently available alternatives to treat malignant bowel obstruction“ (MBO). „Alternative“ in this context means without any surgical intervention. As noted in lines 23-24 „surgery is still the treatment of choice for MBO“ but „50% of patients are unfit for surgery because of poor performance status“. Added to this are high postoperative morbidity and mortality rates in this patient population. As a result, other less invasive measures appear to be more effective. A detailed explanation and presentation of the problem is followed by a brief discussion of drug and surgical treatment options. The less invasive percutaneous or endoscopic treatment options are then presented in detail. This is supplemented with good images. The „expert opinion“ section at the end should be emphasized because this is a useful addition to a pure literature search. In summary, this is a good presentation of the problem and the treatment options of MBO, only the title could perhaps be changed in „An Update on Current Available Alternative Treatments“ because possible surgical measures were only dealt with in 51 lines.

Author Response

As written by the authors in lines 79-80 the aim of the submitted paper “is to give an overview of the currently available alternatives to treat malignant bowel obstruction (MBO)”. “Alternative” in this context means without any surgical intervention. As noted in lines 23-24 “surgery is still the treatment of choice for MBO” but “50% of patients are unfit for surgery because of poor performance status”. Added to this are high postoperative morbidity and mortality rates in this patient population. As a result, other less invasive measures appear to be more effective. A detailed explanation and presentation of the problem is followed by a brief discussion of drug and surgical treatment options. The less invasive percutaneous or endoscopic treatment options are then presented in detail. This is supplemented with good images. The “expert opinion” section at the end should be emphasized because this is a useful addition to a pure literature search. In summary, this is a good presentation of the problem and the treatment options of MBO, only the title could perhaps be changed in “An Update on Current Available Alternative Treatments” because possible surgical measures were only dealt with in 51 lines.

We wish to thank the reviewer for this comment. According to this comment, as among the aims of the paper there was to summarize the currently available treatments for MBO, the section on surgical management has been further detailed as follows: “Most frequently, MSBO is secondary to a condition of diffuse carcinomatosis, for which a more conservative approach is preferable to surgery whenever strangulation of intestinal loops is not present [17]. When needed, possible surgical interventions include small bowel resection with anastomosis, bypass and terminal ostomy [39]. Small bowel ostomy is usually preferred in case of an obstruction site distal enough to allow gastro-intestinal autonomy, thus lowering the risk of short bowel syndrome, or in case of high risk of anastomotic complications [23]. Differently, small bowel resection is suggested in presence of a single carcinomatosis nodule of primitive small bowel malignancy, which however are relative infrequent conditions. MSBO with multiple obstruction sites rep-resents a real challenge for surgeons, due to patients’ frailty, the frequently concomitant state of cachexia and sarcopenia and the advanced stage of the malignancy. For these reasons, intestinal bypass without resection is the preferred treatments in patients with disseminated malignancy, as suggested by a UK prospective study, reporting data that up to 31.9% of patients with MSBO due to carcinomatosis underwent an intestinal bypass [40].

In case of MCO, conservative management is not preferred taking into account the high risk of perforation and death. In the emergency setting such as acute malignant large bowel obstruction, surgery is mandatory, and particularly in patients with right sided colon cancer. The most common surgical procedure is ostomy, including coecostomy or colostomy, which are to be preferred over primary resection and anastomosis or bypass [39,41]. In a multicentre prospective study including 205 patients, the most common intervention in case of right sided colon cancer was resection with ileostomy, that was performed in up to 74.4% of cases [40]. However, in presence of diffuse carcinomatosis, by-pass without resection was the most frequent procedure (31.9%). In patients with single primary tumor obstruction, resection of the tumor should be considered the intervention of choice with regards to higher chances of oncological radicality and longer survival, wherever possible [42].

In contrast, in acute left sided MCO, emergency surgery is not suggested as the treatment of choice. Indeed, the mortality and morbidity rates for emergency surgery in such condition range between 15-34% and 32-64%, respectively, being significantly higher than those of elective surgery [43]. Furthermore, in a recent meta-analysis, emergency surgery with primary anastomosis in patients with left-sided MCO was as-sociated with poorer outcomes, in terms of a higher risk of permanent ostomy, in-tra-hospital morbidity and mortality [44]. Therefore, when possible, strategies allowing to bridge the patient to elective surgery, including transanal drainage or colonic stent placement, should be preferred.” Pages 4 and 5.

Moreover, the expert opinion has been further emphasized (page 10) and, as requested also by reviewer 1, an algorithm summarizing the current available treatments in each situation has been added (Figure 3).